# Efficient High-Order Interaction-Aware Feature Selection Based on Conditional Mutual Information

**Alexander Shishkin, Anastasia Bezzubtseva, Alexey Drutsa,**
**Ilia Shishkov, Ekaterina Gladkikh, Gleb Gusev, Pavel Serdyukov**
Yandex; 16 Leo Tolstoy St., Moscow 119021, Russia
{sisoid,nstbezz,adrutsa,ishfb,kglad,gleb57,pavser}@yandex-team.ru

## Abstract

This study introduces a novel feature selection approach CMICOT, which is a further evolution of filter methods with sequential forward selection (SFS) whose scoring functions are based on conditional mutual information (MI). We state and study a novel saddle point (max-min) optimization problem to build a scoring function that is able to identify joint interactions between several features. This method fills the gap of MI-based SFS techniques with high-order dependencies. In this high-dimensional case, the estimation of MI has prohibitively high sample complexity. We mitigate this cost using a greedy approximation and binary representatives what makes our technique able to be effectively used. The superiority of our approach is demonstrated by comparison with recently proposed interaction-aware filters and several interaction-agnostic state-of-the-art ones on ten publicly available benchmark datasets.

## 1 Introduction

Methods of feature selection is an important topic of machine learning [8, 2, 17], since they improve performance of learning systems while reducing their computational costs. Feature selection methods are usually grouped into three main categories: wrapper, embedded, and filter methods [8]. Filters are computationally cheap and are independent of a particular learning model that make them popular and broadly applicable. In this paper, we focus on most popular filters, which are based on mutual information (MI) and apply the sequential forward selection (SFS) strategy to obtain an optimal subset of features [17]. In such applications as web search, features may be highly relevant only jointly (having a low relevance separately). A challenging task is to account for such interactions [17]. Existing SFS-based filters [18, 3, 24] are able to account for interactions of only up to 3 features.

In this study, we fill the gap in the absence of effective SFS-based filters accounting for feature dependences of higher orders. A search of $t$-way interacting features is turned into *a novel saddle point (max-min) optimization problem* for MI of the target variable and the candidate feature with its *complementary team* conditioned on its *opposing team* of previously selected features. We show that, on the one hand, the saddle value of this conditional MI is a low-dimensional approximation of the CMI score[1] and, on the other hand, solving that problem represents two practical challenges: (a) prohibitively high computational complexity and (b) sample complexity, a larger number of instances required to accurately estimate the MI. These issues are addressed by two novel techniques: (a) *a two stage greedy search* for the approximate solution of the above-mentioned problem whose computational complexity is $O(i)$ at each $i$-th SFS iteration; and (b) *binary representation of features* that reduces the dimension of the space of joint distributions by a factor of $(q/2)^{2t}$ for $q$-value features. Being reasonable and intuitive, these techniques together *constitute the main contribution of our study*: a novel SFS method CMICOT that is able to identify joint interactions between multiple

features. We also empirically validate our approach with 3 state-of-the-art classification models on 10 publicly available benchmark datasets and compare it with known interaction-aware SFS-based filters and several state-of-the-art ones.

## 2 Preliminaries and related work

**Information-theoretic measures.** The *mutual information* (*MI*) of two random variables $f$ and $g$ is defined as $\mathrm{I}(f;g) = \mathrm{H}(f) + \mathrm{H}(g) - \mathrm{H}(f,g)$, where $\mathrm{H}(f) = -\mathrm{E}\left[\log \mathrm{P}(f)\right]$ is Shannon's entropy [4][2]. The *conditional mutual information* of two random variables $f$ and $g$ given the variable $h$ is $\mathrm{I}(f;g \mid h) = \mathrm{I}(f;g,h) - \mathrm{I}(f;h)$. The conditional MI measures the amount of additional information about the variable $f$ carried by $g$ compared to the variable $h$. Given sample data, entropy (and, hence, MI and conditional MI) of discrete variables could be simply estimated using the empirical frequencies (the point estimations) [15] or in a more sophisticated way (e.g., by means of the Bayesian framework [10]). More details on different entropy estimators can be found in [15].

**Background of the feature selection based on MI.** Let $F$ be a set of features that could be used by a classifier to predict a variable $c$ representing a class label. The objective of a feature selection (FS) procedure is to find a feature subset $S^o \subseteq F$ of a given size $k \in \mathbb{N}$ that maximizes its joint MI with the class label $c$, i.e., $S^o = \mathrm{argmax}_{\{S:S \subseteq F, |S| \le k\}} \mathrm{I}(c;S)$. In our paper, we focus on this simple but commonly studied FS objective in the context of MI-based filters [2], though there is a wide variety of other definitions of optimal subset of features [17] (e.g., the all-relevant problem [13]).

In order to avoid an exhaustive search of an optimal subset $S$, most filters are based on sub-optimal search strategies. The most popular one is the *sequential forward selection* (SFS) [20, 23, 17], which starts with an empty set ($S_0 := \varnothing$) and iteratively increases it by adding one currently unselected feature on each step ($S_i := S_{i-1} \cup \{f_i\}, i = 1, \ldots, k$, and $S^o := S_k$). The feature $f_i$ is usually selected by maximizing a certain *scoring function* (also called *score*) $J_i(f)$ that is calculated with respect to currently selected features $S_{i-1}$, i.e., $f_i := \mathrm{argmax}_{f \in F \setminus S_{i-1}} J_i(f)$.

A trivial feature selection approach is to select top-$k$ features in terms of their MI with the class label $c$ [12]. This technique is referred to as MIM [2] and is a particular case of the SFS strategy based on score $J_i^{\mathrm{MIM}}(f) := \mathrm{I}(c;f)$. Note that the resulting set may contain a lot of *redundant* features, since the scoring function $J_i^{\mathrm{MIM}}(\cdot)$ is independent from already selected features $S_{i-1}$. Among methods that take into account the *redundancy* between features [2, 17], the most popular and widely applicable ones are MIFS [1], JMI [21, 14], CMIM [6, 19], and mRMR [16]. Brown et al. [2] unified these techniques under one framework, where they are different low-order approximations of *CMI feature selection approach*. This method is based on the score equal to MI of the label with the evaluated feature conditioned on already selected features:

$$J_i^{\mathrm{CMI}}(f) := \mathrm{I}(c;f \mid S_{i-1}). \tag{1}$$

The main drawback of CMI is the sample complexity, namely, the exponential growth of the dimension of the distribution of the tuple $(c, f, S_{i-1})$ with respect to $i$. The larger the dimension is, the larger number of instances is required to accurately estimate the conditional MI in Eq. (1). Therefore, this technique is not usable in the case of small samples and in the cases, when a large number of features should be selected [2]. This is also observed in our experiment in Appendix.F2, where empirical score estimated over high dimensions results in drastically low performance of CMI.

Thus, low-dimensional approximations of Eq. (1) are more preferable in practice. For instance, the CMIM approach approximates Eq. (1) by

$$J_i^{\mathrm{CMIM}}(f) := \min_{g \in S_{i-1}} \mathrm{I}(c;f \mid g), \tag{2}$$

i.e., one replaces the redundancy of $f$ with respect to the whole subset $S_{i-1}$ by the worst redundancy with respect to one feature from this subset. The other popular methods (mentioned above) are particular cases of the following approximation of the $\mathrm{I}(c;f \mid S_{i-1})$:

$$J_i^{\beta,\gamma}(f) := \mathrm{I}(c;f) - \sum_{g \in S_{i-1}} \left(\beta \mathrm{I}(g;f) - \gamma \mathrm{I}(g;f \mid c)\right), \tag{3}$$

e.g., MIFS ($\beta \in [0,1], \gamma = 0$), mRMR ($\beta = |S_{i-1}|^{-1}, \gamma = 0$), and JMI ($\beta = \gamma = |S_{i-1}|^{-1}$).

An important but usually neglected aspect in FS methods is feature *complementariness* [8, 3] (also known as *synergy* [24] and *interaction* [11]). In general, complementary features are those that appear to have low relevance to the target class $c$ individually, but whose combination is highly relevant [25, 24]. In the next subsection, we provide a brief overview of existing studies on filters that take into account feature interaction. A reader interested in a formalized concept of feature relevance, redundancy, and interaction is referred to [11] and [24].

**Related work on interaction-aware filters.** To the best of our knowledge, existing interaction-aware filters that utilize the pure SFS strategy with a MI-based scoring function are the following ones. RelaxMRMR [18] is a modification of the mRMR method, whose scoring function in Eq. (3) was refined by adding the three-way feature interaction terms $\sum_{h,g \in S_{i-1}, h \neq g} \mathrm{I}(f; h \mid g)$. The method RCDFS [3] is a special case of Eq. (3), where $\beta = \gamma$ are equal to a transformation of the standard deviation of the set $\{\mathrm{I}(f; h)\}_{h \in S_{i-1}}$. The approach IWFS [24] is based on the following idea: at each step $i$, for each unselected feature $f \in F \setminus S_i$, one calculates the next step score $J_{i+1}(f)$ as the current score $J_i(f)$ multiplied by a certain measure of interaction between this feature $f$ and the feature $f_i$ selected at the current step. Both RCDFS and IWFS can catch dependences between no more than 2 features, while RelaxMRMR is able to identify an interaction of up to 3 features, but its score's computational complexity is $O(i^2)$ what makes it unusable in real applications. All these methods could not be straightforwardly improved to incorporate interactions of a higher order.

In our study, we propose a general methodology that fills the gap between the ideal ("oracle") but infeasible CMI method, which takes all interactions into account, and the above-described methods that account for up to 3 interacting features. Our method can be effectively used in practice with its score's computational complexity of a linear growth $O(i)$ (as in most state-of-the-art SFS-filters).

## 3 Proposed feature selection

In this section, we introduce a novel feature selection approach based on the SFS strategy whose score is built by solving from a novel optimization problem and comprises two novel techniques that makes the approach efficient and effective in practice.

### 3.1 Score with $t$-way interacted complementary and opposing teams

Our FS method has a parameter $t \in \mathbb{N}$ that is responsible for the desirable number of features whose mutual interaction (referred to as a *$t$-way feature interaction*) should be taken into account by the scoring function $J_i(\cdot)$. We build the scoring function according to the following intuitions.

First, the amount of relevant information carried by a $t$-way interaction of a candidate feature $f$ has the form $\mathrm{I}(c; f, H)$ for some set of features $H$ of size $|H| \leq t-1$. Second, we remove the redundant part of this information w.r.t. the already selected features $S_{i-1}$ and obtain the non-redundant information part $\mathrm{I}(c; f, H \mid S_{i-1})$. Following the heuristic of the CMIM method, this could be approximated by use of a small subset $G \subseteq S_{i-1}, |G| \leq s \in \mathbb{N}$, i.e., by the low-dimensional approximation $\min_{\{G \subseteq S_{i-1}, |G| \leq s\}} \mathrm{I}(c; f, H \mid G)$ (assuming $s \ll i$). Third, since in the SFS strategy one has to select only one feature at an iteration $i$, this approximated additional information of the candidate $f$ with $H$ w.r.t. $S_{i-1}$ will be gained by with the feature $f$ at this SFS iteration only if all complementary features $H$ have been already selected (i.e., $H \subseteq S_{i-1}$). In this way, the score of the candidate $f$ should be equal to the maximal additional information estimated using above reasoning, i.e., we come to the score which is a solution of the following saddle point (max-min) optimization problem

$$\mathring{J}_i^{(t,s)}(f) := \max_{\substack{H \subseteq S_{i-1}, \\ |H| \leq t-1}} \min_{\substack{G \subseteq S_{i-1}, \\ |G| \leq s}} \mathrm{I}(c; f, H \mid G). \tag{4}$$

We refer to the set $\{f\} \cup H_f^o$, where $H_f^o$ is an optimal set $H$ in Eq. (4), as an *optimal complementary team* of the feature $f \in F \setminus S_{i-1}$, while an optimal set $G$ in Eq. (4) is referred to as an *optimal opposing team* to this feature $f$ (and, thus, to its complementary team as well) and is denoted by $G_f^o$.

The described approach is inspired by methods of greedy learning of ensembles of decision trees [7], where an ensemble of trees is built by sequentially adding a decision tree that maximizes the gain in learning quality. In this way, our complementary team corresponds to the features used in a candidate

decision tree, while our opposing team corresponds to the features used to build previous trees in the ensemble. Since they are already selected by SFS, they are expectedly stronger than $f$ and we can assume that, at the early iterations, a greedy machine learning algorithm would more likely use these features rather than the new feature $f$ once we add it into the feature set. So, Eq. (4) tries to mimic the *maximal amount of information that feature $f$ can provide additionally to the worst-case baseline built on $S_{i-1}$*.

**Statement 1.** *For $t, s + 1 \geq i$, the score $\mathring{J}_i^{(t,s)}$ from Eq. (4) is equal to the score $J_i^{\text{CMI}}$ from Eq. (1).*

The proof's sketch is: (a) justify the identity $\mathring{J}_i^{(t,s)}(f) = \max_{H \subseteq S_{i-1}} \min_{G \subseteq S_{i-1} \setminus H} \text{I}(c; f \mid H, G)$ for $t, s + 1 \geq i$; (b) get a contradiction to the assumption that there are no optimal subsets $H$ and $G$ such that $S_{i-1} = H \cup G$. Detailed proof of Statement 1 is given in Appendix A. Thus, we argue that the score $\mathring{J}_i^{(t,s)}$ from Eq. (4) is a low-dimensional approximation of the CMI score $J_i^{\text{CMI}}$.[3].

The score from Eq. (4) is of a general nature and reasonable, but, to the best of our knowledge, was never considered in existing studies. However, this score is not suitable for effective application, since it suffers from two practical issues:

**(PI.a)** *computational complexity*: efficient search of optimal sets $H_f^o$ and $G_f^o$ in Eq. (4);

**(PI.b)** *sample complexity*: accurate estimation of the MI over features with a large dimension of its joint distribution.

We address these research problems and propose the following solutions to them: in Sec. 3.2, the issue **(PI.a)** is overcome in a greedy fashion, while, in Sec. 3.3, the issue **(PI.b)** is mitigated by means of binary representatives.

## 3.2 Greedy approximation of the score

Note that an exhaustive search of a saddle point in Eq. (4) requires $\binom{i-1}{t-1}\binom{i-1}{s}$ MI calculations that can make calculation of the scoring function $\mathring{J}_i^{(t,s)}$ infeasible at a large iteration $i$ even for low team sizes $t, s > 1$. In order to overcome this issue, we propose the following greedy search for sub-optimal complementary and opposing teams.

At the first stage, we start from a greedy search of a sub-optimal set $H$ that cannot be done straight-forwardly, since Eq. (4) comprises both $\max$ and $\min$ operators. The latter one requires a search of an optimal $G$ that we want do at the second stage (after $H$). Hence, the double optimization problem needs to be replaced by a simpler one which does not utilize a search of $G$.

**Proposition 1.** *(1) For any $H \subseteq S_{i-1}$ such that $|H| \leq s$, the following holds*

$$\min_{G \subseteq S_{i-1}, |G| \leq s} \text{I}(c; f, H \mid G) \leq \text{I}(c; f \mid H). \tag{5}$$

*(2) If $s \geq t - 1$, then the score given by the following optimization problem*

$$\max_{H \subseteq S_{i-1}, |H| \leq t-1} \text{I}(c; f \mid H), \tag{6}$$

*is an upper bound for the score $\mathring{J}_i^{(t,s)}$ from Eq. (4).*

The optimization problem Eq. (6) seems reasonable due to the following properties: (a) in fact, the search of $H$ in Eq. (6) is maximization of the additional information carried out by the candidate $f$ w.r.t. no more than $t - 1$ already selected features from $S_{i-1}$; (b) if a candidate $f$ is a combination of features from $H$, then the right hand side in Eq. (5) is 0 and the inequality becomes an equality.

So, we greedily search the maximum in Eq. (6), obtaining the *(greedy) complementary team* $\{f\} \cup H_f$, where $H_f := \{h_1, \ldots, h_{t-1}\}$ is defined by[4]

$$h_j := \underset{h \in S_{i-1}}{\operatorname{argmax}} \text{I}(c; f \mid h_1, \ldots, h_{j-1}, h), \qquad j = 1, \ldots, t-1. \tag{7}$$

At the second stage, given the complementary team $\{f\} \cup H_f$, we greedily search the *(greedy)* *opposing team* $G_f := \{g_1, \ldots, g_s\}$ in the following way:

$$g_j := \underset{g \in S_{i-1}}{\operatorname{argmin}} \operatorname{I}(c; f, h_1, \ldots, h_{\min\{j,t\}-1} \mid g_1, \ldots, g_{j-1}, g), \qquad j = 1, \ldots, s. \tag{8}$$

Finally, given the teams $\{f\} \cup H_f$ and $G_f$, we get the following greedy approximation of $\overset{\circ}{J}_i^{(t,s)}(f)$:

$$J_i^{(t,s)}(f) := \operatorname{I}(c; f, H_f \mid G_f). \tag{9}$$

This score requires $(t+s-1)i$ MI calculations (see Eq. (7)–(9)), which is a linear dependence on an iteration $i$ as in the most state-of-the-art SFS-based filters [2]. Thus, *we built an efficient approximation of the score $\overset{\circ}{J}_i^{(t,s)}$ and resolve the issue* **(PI.a)**.

Note that we have two options on the minimization stage: either to search among all members of the set $H_f$ at each step (as in Eq. (A.7) in Appendix A.3), or (what we actually do in Eq. (8)) to use only a few first members of $H_f$. The latter option demonstrates noticeably better MAUC performance and also results in 0 score for a feature that is a copy of an already selected one (Proposition 2), while the former does not (Remark A.2 in Appendix A.3). That is why we chose this option.

**Proposition 2.** *Let $s \geq t$ and a candidate feature $f \in F \setminus S_{i-1}$ be such that its copy $\tilde{f} \equiv f$ is already selected $\tilde{f} \in S_{i-1}$, then, in the absence of ties in Eq. (8) for $j \leq t$, the score $J_i^{(t,s)}(f) = 0$.*

Proposition 2 shows that the FS approach based on the greedy score $J_i^{(t,s)}(f)$ remains conservative, i.e., a copy of an already selected feature will not be selected, despite that it exploits sub-optimal teams in contrast to the FS approach based on the optimal score $\overset{\circ}{J}_i^{(t,s)}(f)$.

### 3.3 Binary representatives of features

As it is mentioned in Sec. 2, a FS method that is based on calculation of MI over more than three features is usually not popular in practice, since a large number of features implies a large dimension of their joint distribution that leads to a large number of instances required to accurately estimate the MI [2]. Both our optimal score $\overset{\circ}{J}_i^{(t,s)}$ and our greedy one $J_i^{(t,s)}$ suffer from the same issue **(PI.b)** as well, since they exploit high-dimensional MI in Eq.(4) and Eq. (7)–(9). For instance, if we deal with binary classification and each feature in $F$ has $q$ unique values (e.g., continuous features are usually preprocessed into discrete variables with $q \geq 5$ [18]), then the dimension of the joint distribution of features in Eq. (9) is equal to $2 \cdot q^{t+s}$ (e.g., $\approx 4.9 \cdot 10^8$ for $t = s = 6, q = 5$). In our method, we cannot reduce the number of features used in MIs (since $t$-way interaction constitutes the key basis of our approach), but we can mitigate the effect of the sample complexity by the following novel technique, which we demonstrate on our greedy score $J_i^{(t,s)}$. Let $F$ consists of discrete features[5].

**Definition 1.** For each discrete feature $f \in F$, we denote by $\mathfrak{B}[f]$ the *binary transformation* of $f$, i.e., the set of binary variables (referred to as the *binary representatives* (BR) of $f$) that constitute all together a vector containing the same information as $f$[6]. For any subset $F' \subseteq F$, the set of binary representatives of all features from $F'$ is denoted by $\mathfrak{B}[F'] = \bigcup_{f \in F'} \mathfrak{B}[f]$.

Then, we replace all features by their binary representatives at each stage of our score calculation. Namely, in Eq. (7) and Eq. (8), (a) the searches are performed for each binary representative $b \in \mathfrak{B}[f]$ instead of $f$; (b) the set $H_b^{\texttt{bin}}$ of the complementary team is found among $\mathfrak{B}[S_{i-1}] \cup \mathfrak{B}[f]$; while (c) the opposing team $G_b^{\texttt{bin}}$ is found among $\mathfrak{B}[S_{i-1}]$ (exact formulas could be found in Algorithm 1, lines 12 and 15). Finally, the score of a feature $f$ in this FS approach *based on binary representatives* is defined as the best score among the binary representatives $\mathfrak{B}[f]$ of the candidate $f$:

$$J_i^{(t,s),\texttt{bin}}(f) := \max_{b \in \mathfrak{B}[f]} \operatorname{I}(c; b, H_b^{\texttt{bin}} \mid G_b^{\texttt{bin}}). \tag{10}$$

Note that, in the previous example with a binary target variable $c$ and $q$-value features, the dimension of the joint distribution of binary representatives used to calculate MI in $J_i^{(t,s),\texttt{bin}}$ is equal to $2^{1+t+s}$,

**Algorithm 1** Pseudo-code of the CMICOT feature selection method (an implementation of this algorithm is available at `https://github.com/yandex/CMICOT`).

1: **Input:** $F$ — the set of all features; $\mathfrak{B}[f]$, $f \in F$, — set of binary representatives built on $f$;
2: $c$ — the target variable; $k \in \mathbb{N}$ — the number of features to be selected;
3: $t \in \mathbb{N}, s \in \mathbb{Z}_+$ — the team sizes (parameters of the algorithm);
4: **Output:** $S$ — the set of selected features;
5: **Initialize:**
6: $f_{\texttt{best}} := \operatorname{argmax}_{f \in F} \max_{b \in \mathfrak{B}[f]} \mathrm{I}(c; b);$      *// Select the first feature*
7: $S := \{f_{\texttt{best}}\}; \quad S^{\texttt{bin}} := \mathfrak{B}[f_{\texttt{best}}];$
8: **while** $|S| < k$ **and** $|F \setminus S| > 0$ **do**
9:    **for** $f \in F \setminus S$ **do**
10:      **for** $b \in \mathfrak{B}[f]$ **do**
11:        **for** $j := 1$ **to** $t-1$ **do**
12:          $h_j := \operatorname{argmax}_{h \in S^{\texttt{bin}} \cup \mathfrak{B}[f]} \mathrm{I}(c; b \mid h_1, .., h_{j-1}, h);$   *// Search for complementary feat.*
13:        **end for**
14:        **for** $j := 1$ **to** $s$ **do**
15:          $g_j := \operatorname{argmin}_{g \in S^{\texttt{bin}}} \mathrm{I}(c; b, h_1, .., h_{\min\{j,t\}-1} \mid g_1, .., g_{j-1}, g);$ *// Search for opp. feat.*
16:        **end for**
17:        $J_i[b] := \mathrm{I}(c; b, h_1, .., h_{t-1} \mid g_1, .., g_s);$   *// Calculate the score of the binary rep. b*
18:      **end for**
19:      $J_i[f] := \max_{b \in \mathfrak{B}[f]} J_i[b];$   *// Calculate the score of the feature f*
20:    **end for**
21:    $f_{\texttt{best}} := \operatorname{argmax}_{f \in F \setminus S} J_i[f];$     *// Select the best candidate feature at the current step*
22:    $S := S \cup \{f_{\texttt{best}}\}; \quad S^{\texttt{bin}} := S^{\texttt{bin}} \cup \mathfrak{B}[f_{\texttt{best}}];$
23: **end while**

which is $(q/2)^{t+s}$ times smaller (*the dimension reduction rate*) than for the MI in $J_i^{(t,s)}$. For instance, for $t = s = 6, q = 5$, the MI from Eq. (10) deals with $\approx 8.2 \cdot 10^3$ dimensions, which is $\approx 6 \cdot 10^4$ times lower than $\approx 4.9 \cdot 10^8$ ones for the MI from Eq. (9). The described technique has been inspired by the intuition that probably two binary representatives of two different features interact on average better than two binary representatives of one feature (see App. A.5.1). Therefore, we believe that the BR modification retains the score's awareness to the most interactions between features.

Surely, on the one hand, the BR technique can also be applied to any state-of-the-art SFS-filter [2] or any existing interaction-aware one (RelaxMRMR [18], RCDSFS [3], and IWFS [24]), but the effect on them will not be striking breakthrough, since these filters exploit no more than 3 features in one MI, and the dimension reduction rate will thus be not more than $(q/2)^3$ (e.g., $\approx 15.6$ for $q = 5$). On the other hand, this technique is of a general nature and represents a self-contained contribution to ML community, since it may be applied with noticeable profit to SFS-based filters with MIs of higher orders (possibly not yet invented).

### 3.4 CMICOT feature selection method

We summarize Sec. 3.1–Sec. 3.3 in our novel feature selection method that is based on sequential forward selection strategy with the scoring function from Eq. (10). We refer to this FS method as *CMICOT* (*Conditional Mutual Information with Complementary and Opposing Teams*) and present its pseudo-code in Algorithm 1, which has a form of a SFS strategy with a specific algorithm to calculate the score (lines 10–19). In order to benefit from Prop. 1 and 2, one has to select $s \geq t$, and, for simplicity, from here on in this paper we consider only equally limited teams, i.e., $t = s$.

**Proposition 3.** *Let $|\mathfrak{B}[f]| \leq \nu$, $\forall f \in F$, $|F| \leq M$, and entropies in MIs are calculated over $N$ instances, then $O(i\nu^2 t^2 N)$ simple operations are needed to calculate the score $J_i^{(t,t),\texttt{bin}}$ and $O(k^2 \nu^2 t^2 M N)$ simple operations are needed to select top-$k$ features by CMICOT from Alg. 1.*

Let us remind how each of our techniques contributes to the presented above computational complexity of the score. First, the factor $t^2$ is an expected payment for the ability to be aware of $t$-way interactions (Sec. 3.1). Second, the two stage greedy technique from Sec. 3.2 makes the score' computational complexity linearly depend on a SFS iteration $i$. Third, utilization of the BR technique from Sec. 3.3, on the one hand, seems to increase the computational complexity by the factor $\nu^2$, but, on the other

hand, we know that it drastically reduces the sample complexity (i.e., the number of instances required to accurately estimate the used MIs). For simplicity, let us assume that each feature has $2^\nu$ values and is transformed to $\nu$ binary ones. If we do not use the BR technique, the complexity will be lower by the factor $\nu^2$ for the same number of instances $N$, but estimation of the MIs will require $(2^\nu/2)^{2t}$ times more instances to achieve the same level of accuracy as with the BRs. Hence, the BR technique actually reduces the computational complexity by the factor $2^{2t(\nu-1)}/\nu^2$. Note that the team size $t$ can be used to trade off between the number of instances available in the sample dataset and the maximal number of features whose joint interaction could be taken into account in a SFS manner.

Finally, for a given dataset and a given team size $t$, the score's computational complexity linearly depends on the $i$-th SFS iteration, on the one hand, as in most state-of-the-art SFS-filters [2] like CMIM, MIFS, mRMR, JMI, etc. (see Eq. (2)–(3)). On the other hand, scores of existing interaction-aware ones have either the same ($O(i)$ for RCDFS [3]), or higher ($O(M-i)$ for IWFS [24] and $O(i^2)$ for RelaxMRMR [18]) order of complexity w.r.t. $i$. Thus, we conclude that *our FS method is not inferior in efficiency to all baseline filters, but is able to identify feature dependences of higher orders than these baselines*.

## 4   Experimental evaluation

We compare our CMICOT approach with (a) all known interaction-aware SFS-based filters (RelaxM-RMR [18], IWFS [24], and RCDFS [3]); (b) the state-of-the-art filters [2] (MIFS, mRMR, CMIM, JMI, DISR, and FCBF (CBFS)); (c) and the idealistic but practically infeasible CMI method (see Sec. 2 and [2]). In our experiments, we consider $t = 1, \ldots, 10$ to validate that CMICOT is able to detect interactions of a considerably higher order than its competitors.

**Evaluation on synthetic data.** First, we study the ability to detect high-order feature dependencies using synthetic datasets where relevant and interacting features are a priory known. A synthetic dataset has feature set $F$, which contains a group of jointly interacting relevant features $F_{int}$, and a its target $c$ is a deterministic function of $F_{int}$ for half of examples ($|F \backslash F_{int}| = 15$ and $|F_{int}| = 2, \ldots, 11$ in our experiments). The smaller $k_0 = \min\{k \mid F_{int} \subseteq S_k\}$, the more effective the considered FS method, since it builds the smaller set of features needed to construct the best possible classifier. We conduct an experiment where, first, we randomly sample 100 datasets from the predefined joint distribution (more details in Appendix C). Second, we calculate $k_0$ for each of studied FS methods on these datasets. Finally, we average $k_0$ over the datasets and present the results in Figure 1 (a). We see, first, that CMICOT with $t \geq |F_{int}|$ significantly outperforms all baselines, except the idealistic CMI method whose results are similar to CMICOT. This is expected, since CMI is infeasible only for large $k$, and, in App. F.2, we show that CMICOT is the closest approximation of true CMI among all baselines. Second, the team size $t$ definitely responds to the number of interacted features, that provides an experimental evidence for ability of CMICOT to identify high-order feature interactions.

**Evaluation on benchmark real data.** Following the state-of-the-art practice [6, 22, 2, 18, 24, 3], we conduct an extensive empirical evaluation of the effectiveness of our CMICOT approach on 10 large public datasets from the UCI ML Repo (that include the NIPS'2003 FS competition) and one private dataset from one of the most popular search engines[7]. We employ three state-of-the-art classifiers: Naive Bayes Classifier (NBC), k-Nearest Neighbor (kNN), and AdaBoost [6] (see App. B). Their performance on a set of features is measured by means of AUC [2] (MAUC [9]) for a binary (multi-class) target variable. First, we apply each of the FS methods to select top-$k$ features $S_k$ for each dataset and for $k = 1, .., 50$ [2, 24, 3]. Given $k \in \{1, .., 50\}$, a dataset, and a certain classifier, we measure the performance of a FS method (1) in terms of the (M)AUC of the classifier built on the selected features $S_k$ (2) and in terms of the rank of the FS method among the other FS methods w.r.t. (M)AUC. The resulting (M)AUC and rank averaged over all datasets are shown in Fig. 1(b,c) for kNN and AdaBoost. From these figures we see that our CMICOT for $t = 6$[8] method noticeably outperforms all baselines for the classification models kNN and AdaBoost[9] starting from approximately $k = 10$. We reason this frontier by the size of the teams in CMICOT

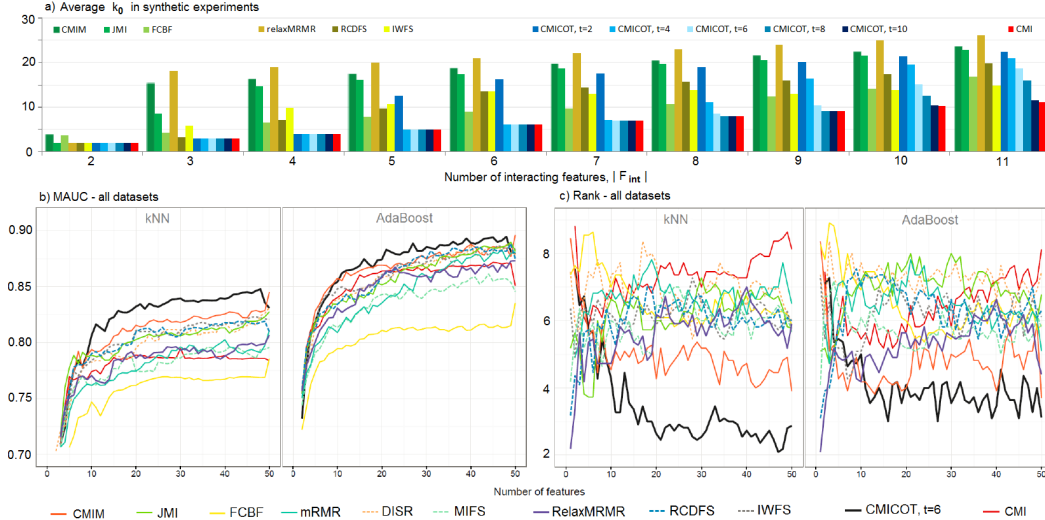

Figure 1: (a) Comparison of the performance of SFS-based filters in terms of average $k_0$ on synthetic datasets. (b) Average values of (M)AUC for compared FS methods and (c) their ranks w.r.t. (M)AUC $k = 1, .., 50$ and for the kNN and AdaBoost classification models over all datasets (see also App. C,E).

method, which should select different teams more likely when $|S_{i-1}| > 2t (= 12$ for $t = 6)$. The curves on Fig. 1 (b,c) are obtained over a test set, while a 10-fold cross-validation [2, 18] is also applied for several key points (e.g. $k = 10, 20, 50$) to estimate the significance of differences in classification quality. The detailed results of this CV for $k = 50$ on representative datasets are given in Appendix E.2. A more comprehensive details on these and other experiments are in App. E and F.

We find that our approach either significantly outperforms baselines (most one for kNN and AdaBoost), or have non-significantly different difference with the other (most one for NBC). Note that the interaction awareness of RelaxMRMR, RCDFS and IWFS is apparently not enough to outperform CMIM, our strongest competitor. In fact, there is no comparison of RelaxMRMR and IWFS with CMIM in [3, 24], while RCDFS is outperformed by CMIM on some datasets including the only one utilized in both [18] and our work. One compares CMICOT with and without BR technique: on the one hand, we observed that CMICOT without BRs loses in performance to the one with BRs on the datasets with non-binary features, that emphasizes importance of the problem **(PI.b)**; on the other hand, results on binary datasets (poker, ranking, and semeion; see App. E), where the CMICOT variants are the same, the effectiveness of our approach separately to the BR technique is established.

# 5 Conclusions

We proposed a novel feature selection method CMICOT that is based on sequential forward selection and is able to identify high-order feature interactions. The technique based on a two stage greedy search and binary representatives of features makes our approach able to be effectively used on datasets of different sizes for restricted team sized $t$. We also empirically validated our approach for $t$ up to 10 by means of 3 state-of-the-art classification models (NBC, kNN, and AdaBoost) on 10 publicly available benchmark datasets and compared it with known interaction-aware SFS-based filters (RelaxMRMR, IWFS, and RCDFS) and several state-of-the-art ones (CMIM, JMI, CBFS, and others). We conclude that our FS algorithm, unlike all competitor methods, is capable to detect interactions between up to $t$ features. The overall performance of our algorithm is the best among the state-of-the-art competitors.

## Acknowledgments

We are grateful to Mikhail Parakhin for important remarks which resulted in significant improvement of the paper presentation.

---

RelaxMRMR also showed its poorest performance on NBC in [18], while IWFS and RCDFS in [3, 24] didn't consider NBC at all.

## Footnotes

[1]The CMI filter is believed to be a "north star" for vast majority of the state-of-the-art filters [2].

[2]From here on in the paper, variables separated by commas or a set of variables in MI expressions are treated as one random vector variable, e.g., $\mathrm{I}(f;g,h) := \mathrm{I}\left(f;(g,h)\right)$ and, for $F = \cup_{i=1}^n \{f_i\}$, $\mathrm{I}(f;F) := \mathrm{I}(f;f_1,..,f_n)$.

[3]Moreover, the CMIM score from Eq. (2) is a special case of Eq. (4) with $s = t = 1$ and restriction $G \neq \varnothing$.

[4]If several elements provide an optimum (the case of ties), then we randomly select one of them.

[5]If there is a non-discrete feature, then we apply a discretization (e.g., by equal-width, equal-frequency binnings [5], MDL [22, 3], etc.), which is the state-of-the-art preprocessing of continuous features in filters.

[6]For instance, for $f$ with values in $\{x_l\}_{l=1}^q$, one could take $\mathfrak{B}[f] = \{I_{\{f=x_l\}}\}_{l=1}^{q-1}$, where $I_{\mathcal{X}}$ is $\mathcal{X}$'s indicator, or take bits of a binary encoding of $\{x_l\}_{l=1}^q$ that is a smallest set (i.e., $|\mathfrak{B}[f]| = \lceil \log_2 q \rceil$) among possible $\mathfrak{B}[f]$.

[7]The number of features, instances, and target classes varies from 85 to 5000, from 452 to $10^5$, and from 2 to 26 respectively. More datasets' characteristics and preprocessing can be found in Appendix D.

[8]Our experimentation on CMICOT with different $t = 1, \ldots, 10$ on our datasets showed that $t = 5$ and 6 are the most reasonable in terms of classifier performance (see Appendix E.1.1).

[9]The results of CMICOT on NBC classifier are similar to the ones of other baselines. This is expected since NBC does not exploit high-order feature dependences, which is the key advantage of CMICOT. Note that

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
