[Supplementary Material · 2016-nips-fsmi-supp-final.pdf]

# Efficient High-Order Interaction-Aware Feature Selection Based on Conditional Mutual Information SUPPLEMENTARY MATERIAL

**Alexander Shishkin, Anastasia Bezzubtseva, Alexey Drutsa,**
**Ilia Shishkov, Ekaterina Gladkikh, Gleb Gusev, Pavel Serdyukov**
Yandex; 16 Leo Tolstoy St., Moscow 119021, Russia
{sisoid,nstbezz,adrutsa,ishfb,kglad,gleb57,pavser}@yandex-team.ru

## Contents

# A Missing proofs and intuition details from Section 3

## A.1 Proof of Statement 1

*Proof.* The team sizes $t, s + 1 \geq i$ imply that the search of the optimal sets $H$ and $G$ in max/min operators of the score $\mathring{J}_i^{(t,s)}$ is made over any subset of the already selected features $S_{i-1}$ (since its size $|S_{i-1}| = i - 1$):

$$\mathring{J}_i^{(t,s)}(f) = \max_{H \subseteq S_{i-1}} \min_{G \subseteq S_{i-1}} \mathrm{I}(c; f, H \mid G),$$

i.e., there is no actual limitation on the sizes $|H|$ and $|G|$.

For any candidate $f \in F \setminus S_{i-1}$ and any subsets $H, G \subseteq S_{i-1}$, on the one hand, the following inequality holds

$$\mathrm{I}(c; f, H|G) \geq \mathrm{I}(c; f, H|H, G) = \mathrm{I}(c; f|H, G).$$

On the other hand, denoting $G_1 = H \cup G$,

$$\mathrm{I}(c; f|H, G) = \mathrm{I}(c; f, H|H, G) = \mathrm{I}(c; f, H|G_1) \geq \min_{G' \subseteq S_{i-1}} \mathrm{I}(c; f, H \mid G').$$

Therefore, for any $H \subseteq S_{i-1}$, one has

$$\min_{G \subseteq S_{i-1}} \mathrm{I}(c; f, H \mid G) = \min_{G \subseteq S_{i-1}} \mathrm{I}(c; f \mid H, G) = \min_{G \subseteq S_{i-1} \setminus H} \mathrm{I}(c; f \mid H, G).$$

Finally, for the score $\mathring{J}_i^{(t,s)}$, we get

$$\mathring{J}_i^{(t,s)}(f) = \max_{H \subseteq S_{i-1}} \min_{G \subseteq S_{i-1} \setminus H} \mathrm{I}(c; f \mid H, G). \tag{A.1}$$

Let $H_0$ be an optimal set $H$ from Eq. (A.1) with *the largest size* and let $G_0$ be an optimal set $G$ from Eq. (A.1) w.r.t. $H_0$. Note that $\mathring{J}_i^{(t,s)}(f) = \mathrm{I}(c; f \mid H_0, G_0)$. We will prove that, in this case, $H_0 \cup G_0 = S_{i-1}$. To prove this, let us assume the contrary, namely, that there is $w \in S_{i-1} \setminus (H_0 \cup G_0)$. On the one hand, by the definition of $H_0$, one has

$$\min_{G \subseteq S_{i-1} \setminus (H_0 \cup \{w\})} \mathrm{I}(c; f \mid H_0, w, G) < \mathrm{I}(c; f \mid H_0, G_0), \tag{A.2}$$

where inequality is strong, since $H_0 \cup \{w\}$ is larger than $H_0$ and, thus, could not be an optimal set $H$ in Eq. (A.1). On the other hand, by the definition of $G_0$,

$$\mathrm{I}(c; f \mid H_0, G_0) \leq \mathrm{I}(c; f \mid H_0, w, G) \qquad \forall G \subseteq S_{i-1}. \tag{A.3}$$

Combining Eq. (A.2) and Eq. (A.3), we get a contradiction. So, the right hand side of Eq. (A.1) is equal to $\mathrm{I}(c; f \mid S_{i-1})$, which is the score of CMI. $\square$

## A.2 Proof of Proposition 1

*Proof.* Let us demonstrate that the first statement holds. Let $H \subseteq S_{i-1}$ be such that $|H| \leq s$, then, on the one hand,

$$\min_{G \subseteq S_{i-1}, |G| \leq s} \mathrm{I}(c; f, H \mid G) \leq \mathrm{I}(c; f, H \mid H), \tag{A.4}$$

since $H$ is in the set over which the minimum is taken. On the other hand,

$$\mathrm{I}(c; f, H \mid H) = \mathrm{I}(c; f, H, H) - \mathrm{I}(c; H) = \mathrm{I}(c; f, H) - \mathrm{I}(c; H) = \mathrm{I}(c; f \mid H). \tag{A.5}$$

Combining Eq. (A.4) and (A.5), we get the first part of the proposition. For $t - 1 \leq s$, this implies the second statement of the proposition, since, for an optimal complimentary team $\{f\} \cup H_f^o$, one has

$$\mathring{J}_i^{(t,s)}(f) = \min_{G \subseteq S_{i-1}, |G| \leq s} \mathrm{I}(c; f, H_f^o \mid G) \leq \mathrm{I}(c; f \mid H_f^o) \leq \max_{H \subseteq S_{i-1}, |H| \leq t-1} \mathrm{I}(c; f \mid H).$$

$\square$

### A.3   Proof of Proposition 2 and discussion

First, we prove Proposition 2. Second, we show that the absence of ties is a necessary condition for the validity of the statement of Proposition 2 (see Remark A.1). Finally, we demonstrate that a statement similar to Proposition 2 , but for an alternative search of greedy opposing team does not hold (see Remark A.2).

*Proof of Proposition 2.*  Let us consider the first step of the search of the greedy opposing team $G_f$, i.e., $j = 1$. Then, since $\tilde{f} \in S_{i-1}$,

$$0 \le \min_{g \in S_{i-1}} \mathrm{I}(c; f \mid g) \le \mathrm{I}(c; f \mid \tilde{f}) = \mathrm{I}(c; f \mid f) = 0.$$

Hence, the equality $\mathrm{I}(c; f \mid g_1) = 0$ holds for $g_1$ as well. Since there are no ties (i.e., the minimum is unique), $g_1 = \tilde{f}$.

Then, we proceed by induction. We consider an arbitrary step $j$ of the search of the greedy opposing team $G_f$ s.t. $1 < j \le t$, and we assume that

$$\mathrm{I}(c; f, h_1, \ldots, h_{j-2} \mid g_1, \ldots g_{j-1}) = 0,$$

with $g_1 = \tilde{f}$ and $g_l = h_{l-1}, l = 2, \ldots, j-1$. In this case, we search for

$$g_j = \operatorname*{argmin}_{g \in S_{i-1}} \mathrm{I}(c; f, h_1, \ldots, h_{j-1} \mid \tilde{f}, h_1, \ldots, h_{j-2}, g).$$

Since $h_{j-1} \in S_{i-1}$,

$$0 \le \min_{g \in S_{i-1}} \mathrm{I}(c; f, h_1, \ldots, h_{j-1} \mid \tilde{f}, h_1, \ldots, h_{j-2}, g) \le \mathrm{I}(c; f, h_1, \ldots, h_{j-1} \mid \tilde{f}, h_1, \ldots, h_{j-2}, h_{j-1})$$

$$= \mathrm{I}(c; f, h_1, \ldots, h_{j-1} \mid f, h_1, \ldots, h_{j-2}, h_{j-1}) = 0$$

Due to the absence of ties, we get $g_j = h_{j-1}$.

So, for the case, when $s = t$, we have

$$J_i^{(t,s)}(f) = \mathrm{I}(c; f, h_1, \ldots, h_{t-1} \mid \tilde{f}, h_1, \ldots, h_{t-1}) = 0. \tag{A.6}$$

Finally, let us consider the case $s > t$. Then again, we proceed by induction, where Eq. (A.6) is the basis for this induction. We assume that for a step $j > t$ of the search of the greedy opposing team $G_f$, the following holds

$$\mathrm{I}(c; f, h_1, \ldots, h_{t-1} \mid g_1, \ldots, g_{j-1}) = 0,$$

with $g_1 = \tilde{f}$ and $g_l = h_{l-1}, l = 2, \ldots, t$.

Then,

$$\min_{g \in S_{i-1}} \mathrm{I}(c; f, h_1, \ldots, h_{t-1} \mid g_1, \ldots, g_{j-1}, g) \le \mathrm{I}(c; f, h_1, \ldots, h_{t-1} \mid g_1, \ldots, g_{j-1}, g_1)$$

$$= \mathrm{I}(c; f, h_1, \ldots, h_{t-1} \mid g_1, \ldots, g_{j-1}) = 0.$$

Hence, the equality $\mathrm{I}(c; f, h_1, \ldots, h_{t-1} \mid g_1, \ldots, g_{j-1}, g) = 0$ holds for any optimal $g = g_j$ as well. So, for the case, when $s > t$, we have

$$J_i^{(t,s)}(f) = \mathrm{I}(c; f, h_1, \ldots, h_{t-1} \mid \tilde{f}, h_1, \ldots, h_{t-1}, g_{t+1}, \ldots, g_s) = 0.$$

$\square$

**Remark A.1.** The statement of Proposition 2 is invalid in the case when a tie exists. Namely, if, during the greedy search of the opposing team, there is a step $j \le t$ at which there are multiple minimums, and the opposer $g_j$ is randomly selected among these minimums, then a copy $f \in F \setminus S_{i-1}$ of an already selected feature $\tilde{f} \in S_{i-1}$ might get a non-zero score $J_i^{(t,s)}(f)$.

*Proof.* Consider $t = s = 2$, features $f_0, f_1, f_2, f_3$ and a target $c$, such that $f_0 = f_1$,

$$\mathrm{I}(c; f_0) = \mathrm{I}(c; f_1) \leq \varepsilon, \quad \mathrm{I}(c; f_2) \leq \varepsilon, \quad \mathrm{I}(c; f_3) = \mathrm{I}(c; f_1, f_3) \leq \varepsilon, \quad \mathrm{I}(c; f_2, f_3) \leq \varepsilon,$$

but

$$\mathrm{I}(c; f_1, f_2) = \mathrm{H}(c) \gg \varepsilon.$$

Random variables with such properties could be constructed by means of proper Bernoulli distributions[1].

Let us consider the state $i = 3$, where $S_{i-1} = \{f_1, f_2, f_3\}$[2], and we calculate the score $J_i^{(t,s)}(f_0)$. Then, during the greedy search of $H_{f_0}$, one has $h_1 := f_2$. During the greedy search of $G_{f_0}$, at the first step $j = 1$, one obtains that, for $g = f_1$,

$$\mathrm{I}(c; f_0 \mid f_1) = 0,$$

and, thus,

$$\min_{g \in S_{i-1}} \mathrm{I}(c; f_0 \mid g) = 0.$$

But, for $g = f_3$,

$$\mathrm{I}(c; f_0 \mid f_3) = \mathrm{I}(c; f_0, f_3) - \mathrm{I}(c; f_3) = \varepsilon - \varepsilon = 0$$

as well. Let us assume that the algorithm sets $g_1 := f_3$ (since ties are resolved randomly). At the next step $j = 2$ of the greedy search of $G_{f_0}$, one has, for $g = f_1$:

$$\mathrm{I}(c; f_0, f_2 \mid f_3, f_1) = \mathrm{I}(c; f_0, f_2, f_3, f_1) - \mathrm{I}(c; f_3, f_1) \geq \mathrm{H}(c) - \varepsilon > 0,$$

for $g = f_2$:

$$\mathrm{I}(c; f_0, f_2 \mid f_3, f_2) = \mathrm{I}(c; f_0, f_2, f_3, f_2) - \mathrm{I}(c; f_3, f_2) \geq \mathrm{H}(c) - \varepsilon > 0,$$

and, for $g = f_3$:

$$\mathrm{I}(c; f_0, f_2 \mid f_3, f_3) = \mathrm{I}(c; f_0, f_2, f_3, f_3) - \mathrm{I}(c; f_3, f_3) \geq \mathrm{H}(c) - \varepsilon > 0.$$

Therefore,

$$J_i^{(t,s)}(f_0) = \min_{g \in S_{i-1}} \mathrm{I}(c; f_0, h_1 \mid g_1, g) > 0.$$

$\square$

**Remark A.2.** The statement of Proposition 1 does not hold for the alternative greedy search strategy, in which the alternative greedy opposing team $\hat{G}_f := \{\hat{g}_1, \ldots, \hat{g}_s\}$ is built by:

$$\hat{g}_j := \operatorname*{argmin}_{g \in S_{i-1}} \mathrm{I}(c; f, H_f \mid \hat{g}_1, \ldots, \hat{g}_{j-1}, g), \qquad j = 1, \ldots, s, \tag{A.7}$$

where $\{f\} \cup H_f$ is the greedy complimentary team, defined in Section 3.2.

*Proof.* Let us consider $t = s = 2$, features $f_0, f_1, f_2, f_3$, and a target $c$ such that $f_0 = f_1$,

$$\mathrm{I}(c; f_0) = \mathrm{I}(c; f_1) \leq \varepsilon, \quad \mathrm{I}(c; f_2) \leq \varepsilon, \quad \mathrm{I}(c; f_3) = \frac{1}{2}\mathrm{H}(c),$$

$$\mathrm{I}(c; f_2, f_3) \leq \frac{1}{2}\mathrm{H}(c) + \varepsilon, \quad \mathrm{I}(c; f_1, f_3) \leq \frac{1}{2}\mathrm{H}(c) + \varepsilon, \quad \mathrm{I}(c; f_1, f_2) = \mathrm{H}(c) \gg 2\varepsilon.$$

Random variables with such properties could be constructed by means of proper Bernoulli distributions in a similar way as in the proof of the previous remark.

Let us consider the state $i = 3$, where $S_{i-1} = \{f_1, f_2, f_3\}$, and we calculate the modification of the score $J_i^{(t,s)}(f_0)$ that is defined by:

$$\widehat{J}_i^{(t,s)}(f) := \mathrm{I}(c; f, H_f \mid \hat{G}_f), \tag{A.8}$$

where $\hat{G}_f$ is constructed via Eq. (A.7).

Then, during the greedy search of $H_{f_0}$, one has $h_1 := f_2$. During the greedy search of $\hat{G}_{f_0}$, at the first step $j = 1$, one obtains that, for $g = f_1$,

$$\mathrm{I}(c; f_0, h_1 \mid f_1) = \mathrm{I}(c; f_0, f_2 \mid f_0) = \mathrm{I}(c; f_0, f_2) - \mathrm{I}(c; f_0) \geq \mathrm{H}(c) - \varepsilon,$$

for $g = f_2$,

$$\mathrm{I}(c; f_0, h_1 \mid f_2) = \mathrm{I}(c; f_0, f_2 \mid f_2) = \mathrm{I}(c; f_0, f_2) - \mathrm{I}(c; f_2) \geq \mathrm{H}(c) - \varepsilon,$$

and, for $g = f_3$,

$$\mathrm{I}(c; f_0, h_1 \mid f_3) = \mathrm{I}(c; f_0, f_2 \mid f_3) = \mathrm{I}(c; f_0, f_2, f_3) - \mathrm{I}(c; f_3) = \mathrm{H}(c) - \frac{1}{2}\mathrm{H}(c),$$

Since, $\mathrm{H}(c) > 2\varepsilon$, the algorithm selects the unique minimum: $\hat{g}_1 := f_3$. At the next step $j = 2$ the greedy search of $G_{f_0}$, one has, for $g = f_1$:

$$\mathrm{I}(c; f_0, f_2 \mid f_3, f_1) = \mathrm{I}(c; f_0, f_2, f_3, f_1) - \mathrm{I}(c; f_3, f_1) \geq \mathrm{H}(c) - \frac{1}{2}\mathrm{H}(c) - \varepsilon > 0,$$

for $g = f_2$:

$$\mathrm{I}(c; f_0, f_2 \mid f_3, f_2) = \mathrm{I}(c; f_0, f_2, f_3, f_2) - \mathrm{I}(c; f_3, f_2) \geq \mathrm{H}(c) - \frac{1}{2}\mathrm{H}(c) - \varepsilon > 0,$$

for $g = f_3$:

$$\mathrm{I}(c; f_0, f_2 \mid f_3, f_3) = \mathrm{I}(c; f_0, f_2, f_3, f_3) - \mathrm{I}(c; f_3, f_3) \geq \mathrm{H}(c) - \frac{1}{2}\mathrm{H}(c) > 0.$$

Therefore,

$$\widehat{J}_i^{(t,s)}(f_0) = \min_{g \in S_{i-1}} \mathrm{I}(c; f_0, h_1 \mid \hat{g}_1, g) > 0.$$

$\square$

### A.4   Proof of Proposition 3

*Proof.* The calculation of a joint entropy of $m$ variables over $N$ instances (samples) takes $O(mN)$ simple operations. Indeed, to compute the empirical entropy $H(F)$ of a feature vector $F$ of size $m$, we group all instances in the dataset by the observed values of the full feature vector $F$, this takes $O(Nm)$ operations. Then, for each observed value $X$, we compute $-(N_X/N)\log(N_X/N)$, where $N_X$ is the number of instances with $F = X$. This takes not greater than $O(Nm)$ operations. At last, we sum up the obtained values. Hence, any MI that involve $m$ variables requires $O(mN)$ simple operations as well.

Let us estimate the number of MIs that are calculated by our algorithm at the $i$-th SFS iteration for each candidate feature $f \in F \setminus S_{i-1}$. So, the search of the complementary team is done via $t - 1$ steps (lines 11-13 in Alg. 1), where, at each step $j$, one finds a maximal MI with $2 + j$ variables over $\mathfrak{B}[S_{i-1}] \cup \mathfrak{B}[f]$ (line 12 in Alg. 1) . Hence, the search of the complementary team no more than $(t-1)i\nu$ times calls a calculation of MIs that involve no more than $t + 1$ variables. Similarly, the search of the opposing team (lines 14-16 in Alg. 1) no more than $t(i-1)\nu$ times calls a calculation of MIs that involve no more than $2t + 1$ variables.

These teams are found for each binary representative (lines 10-17 in Alg. 1) of the candidate $f$. Thus, the score $J_i^{(t,t),\texttt{bin}}(f)$ requires calculation of $\big((t-1)i + t(i-1)\big)\nu^2$ of MIs that involve no more than $2t + 1$ variables . Thus, the overall score requires $\big((t-1)i + t(i-1)\big)\nu^2 O\big((2t+1)N\big) = O(i\nu^2 t^2 N)$ simple operations. Thus, we proved the first part of the proposition.

In order to estimate the required number of simple operations to select top-$k$ features by CMICOT, note that the score $J_i^{(t,t),\texttt{bin}}$ is calculated $M - i + 1$ times (over the features $F \setminus S_{i-1}$) at the $i$-th SFS iteration (lines 9-19 in Alg. 1). Since we run $k$ SFS iterations, the overall computation complexity is:

$$O(1) + \sum_{i=1}^{k}(M - i + 1)O(i\nu^2 t^2 N) = O(k^2\nu^2 t^2 MN).$$

$\square$

### A.5  Details on some intuitions

#### A.5.1  Two binary representatives of one feature vs. the ones of two different features

Our technique of binary representatives has been inspired by the intuition that two binary representatives of two different features interact probably on average better than two binary representatives of one feature.

To clarify this, let us consider a dataset of vectors whose components are $2^q$-valued features and assume that the dataset size allows to estimate the entropy of a vector whose domain of values does not exceed $2^{qr}$ to achieve a required accuracy level. Let us represent each feature by $q$ binary ones. Without BR technique (a), interactions are searched among only $r$ $q$-tuple combinations, while with BR technique (b), we are able to combine any $qr$ binary features. In the case (b), the algorithm has more freedom to find high interactions than in the case (a), since the variants in (a) constitutes a subset of the variants in (b).

## B  Information on the realizations of baseline methods and employed classifiers

We utilized the following realizations of our baselines

- the state-of-the-art filters MIFS, mRMR, CMIM, JMI, DISR, FCBF (CBFS), and CMI Brown et al. (2012): the implementation provided by the FEAST Toolbox from `http://www.cs.man.ac.uk/~gbrown/fstoolbox/`;
- RelaxMRMR: the Matlab implementation provided by the authors Vinh et al. (2015);
- IWFS and RCDFS: ours Matlab implementation.

We employ three state-of-the-art classifiers:

- Naive Bayes Classifier (NBC) used in Novovičová et al. (2007); Vergara and Estévez (2010); Tsimpiris et al. (2012);
- k-Nearest Neighbor (kNN) used in Fleuret (2004); Meyer et al. (2008); Lee et al. (2012): k is set to 3;
- AdaBoost used in Fleuret (2004): 200 ensemble members in SAMME multi-class AdaBoost with CART trees of maximum depth 6 as base learners (scikit-learn implementation).

## C  Details on the synthetic experiments

### C.1  The synthetic dataset setup

In our synthetic experiments, we construct our dataset in the following way. Given the number of desired interacting features $n \geq 2$ (a parameter of the dataset), we construct a set $F$ of $n+15$ features, which contains a group of jointly interacting relevant features $F_{int} = n$, and a part of the target $c$ is a deterministic function of $F_{int}$.

So, first of all, let us consider some auxiliary independent random variables:

- 3 jointly i.i.d. Bernoulli random variables $u, \xi, \theta$ with success probability $1/2$.
- 1 uniform random variable $\Xi$ over the set $\mathbb{Z}_{n-1}$;
- 1 uniform random variable $\Theta$ over the set $\mathbb{Z}_{10}$;
- 5 jointly i.i.d. Bernoulli random variables $\epsilon_1, \ldots, \epsilon_5$ with success probability $1/2$.
- overall, the variables $u, \Xi, \Theta, \xi, \theta, \epsilon_1, \ldots, \epsilon_5$ are jointly independent as well.

Based on these variables we define the following variables (let $I_A$ be the indicator of the event $A$):

- $n-1$ variables:

$$w_1 := u \cdot \xi \cdot I_{\{\Xi=0\}}, \quad \ldots, \quad w_{n-1} := u \cdot \xi \cdot I_{\{\Xi=n-2\}};$$

- 10 variables:

$$v_1 := u \cdot \theta \cdot I_{\{\Theta=0\}}, \quad \ldots, \quad v_{10} := u \cdot \theta \cdot I_{\{\Theta=9\}};$$

Then, we define:

- The target:

$$c := u$$

- $n$ relevant and interacting features:

$$F_{int} := \{u, w_1, \ldots, w_{n-1}\};$$

- 10 relevant but non-interacting features (each of them (as individual feature) behaves similarly as one of $w_1, \ldots, w_{n-1}$):

$$F_{rel-not-int} := \{v_1, \ldots, v_{10}\};$$

- 5 irrelevant features:

$$F_{irr} := \{\epsilon_1, \ldots, \epsilon_5\}.$$

Thus, our set consists of

$$F = F_{int} \cup F_{rel-not-int} \cup F_{irr}.$$

First, note that $w_1 + \ldots + w_{n-1} = c \cdot \xi$ and $v_1 + \ldots + v_{10} = c \cdot \theta$. Hence, knowing $w_1, \ldots, w_{n-1}$, and $\xi$, one can restore the target on a half of instances (those where $\xi = 1$), i.e. these features constitutes a $n$-way interaction. But one cannot do a similar derivation for $v_1, \ldots, v_{10}$ since the variable $\theta$ is not included in our set of features $F$. Second, note that the feature $\xi$ is irrelevant (i.e., behave similarly as the irrelevant features $\epsilon_1, \ldots, \epsilon_5$) until it is considered without the features $w_1, \ldots, w_{n-1}$.

Hence, in this synthetic experiment, a FS method should select the features $F_{int}$ as earlier as possible from all $15 + n$ features $F$.

In our experiments, for each $n = 2, \ldots, 11$, we randomly sample 100 datasets from the predefined joint distribution with 1000 instances.

## C.2 Details on the results of the synthetic experiments

The performance of a feature selection method on the synthetic datasets is measured in terms of

$$k_0 = \min\{k \mid F_{int} \subseteq S_k\}.$$

The smaller $k_0$, the more effective the considered FS method, since it builds the smaller set of features needed to construct the best classifier. Hence, for each predefined joint distribution ($n = 2, \ldots, 11$), for each of 100 random sample, we calculate $k_0$ for each of studied FS methods on these datasets (namely, 6 the state-of-the-art filters Brown et al. (2012) (MIFS, mRMR, CMIM, JMI, DISR, and FCBF (CBFS)); all known interaction-aware SFS-based filters (RelaxMRMR Vinh et al. (2015), IWFS Zeng et al. (2015), and RCDFS Chen et al. (2015)); the CMI method; and our method CMICOT with $t = 2, 4, 6, 8, 10$). Finally, we average $k_0$ over the datasets and present the results in Figure C.1.

## D Description of the used benchmark datasets

In Table D, we present descriptive statistics of 10 public datasets collected from the UCI Machine Learning Repository[3] and one private dataset of a learning-to-rank problem obtained from one of the most popular search engines (listed as "ranking"). We preprocessed the data for our needs (i.e., removed index and nominal columns, discretized continuous target variables, replaced missing values). Details on the dataset manipulations could be find in Table D. The datasets represent both binary classification and multi-classification (up to 26 classes) problems. Some of the datasets have solely binary features (including target feature), while datasets with both discrete and continuous features are also presented. We discretized features with more than 10 unique values into 10-value discrete ones using an equal-width binning Dougherty et al. (1995); Brown et al. (2012). The resulting median number of bins for each dataset is presented in Table D, as

Figure C.1: Comparison of the performance of SFS-based filters in terms of the average $k_0$ on synthetic datasets.

well as dataset difficulty ratios (smaller values indicate more complicated tasks) Vinh et al. (2015).

Table D.1: Dataset description (types of features: binary (b), discrete (d), continuous (c)).

| Dataset | Instances $(N)$ | Features $(M)$ | Classes | Median arity | Ratio $(N/M)$ | Type of features |
|---|---|---|---|---|---|---|
| arrhythmia[1] | 452 | 279 | 13 | 8 | 1.6 | (b) (d) (c) |
| coil2000 | 9822 | 85 | 8 | 10 | 115.6 | (b) (d) (c) |
| gizette | 6000 | 5000 | 2 | 10 | 1.2 | (b) (d) (c) |
| isolet | 7797 | 617 | 26 | 10 | 12.6 | (c) |
| libras | 27936 | 300 | 2 | 9 | 93.1 | (c) |
| madelon | 2000 | 500 | 2 | 10 | 4.0 | (d) (c) |
| poker[2] | 100000 | 110 | 2 | 2 | 909.1 | (b) |
| ranking[3] | 50000 | 117 | 2 | 2 | 427.4 | (b) |
| semeion | 1593 | 256 | 10 | 2 | 6.2 | (b) |
| smartphone | 10929 | 561 | 12 | 10 | 19.5 | (c) |
| usps | 9298 | 256 | 10 | 10 | 36.3 | (c) |

[1] missing values were replaced with 0s.
[2] sample 100000 rows of the dataset were used, the original 10 categorical features were transformed into 110 binary features using equal-width and equal-frequency strategies, label was binarized ('0' was assigned to target value 0, '1' was assigned to target values greater than 0).

[3] a private dataset obtained from one of the most popular search engines.

# E  Additional information on classification quality experiments

## E.1  Aggregated results

### E.1.1  Aggregated results over all datasets for each $k$

The overall performance of a method can be presented as a mean (M)AUC averaged over $k = 1..50$. Yet again, CMICOT is better than all the competitors. For the sake of a better comprehension the (M)AUC values can be transformed to rankings on each $k$ (the method with the highest value has rank 1, the worst method has rank 11). The averaged rankings confirm the previous conclusions. Particularly, on kNN CMICOT's rank is 71% higher than that of the closest competitor, CMIM.

The average values of (M)AUC for $k = 1, \ldots, 50$ and for each of three classification models (NBC, kNN, and AdaBoost) over all datasets are presented for CMICOT with different $t = 1, \ldots, 10$ in Fig. E.1, while its ($t = 6$) comparison with baselines in Fig. E.2.

The average ranks of compared FS methods w.r.t. (M)AUC for $k = 1, \ldots, 50$ and for each of three classification models (NBC, kNN, and AdaBoost) over all datasets are presented for CMICOT with different $t = 1, \ldots, 10$ in Fig. E.3, while its ($t = 6$) comparison with baselines in Fig. E.4.

Figure E.1: The average values of (M)AUC for $k = 1, \ldots, 50$ and for each of three classification models (NBC, kNN, and AdaBoost) over all datasets for CMICOT with different $t = 1, \ldots, 10$.

Figure E.2: The average values of (M)AUC for $k = 1, \ldots, 50$ and for each of three classification models (NBC, kNN, and AdaBoost) over all datasets for all baselines and CMICOT with $t = 6$.

Figure E.3: The average ranks of compared FS methods w.r.t. (M)AUC for $k = 1, \ldots, 50$ and for each of three classification models (NBC, kNN, and AdaBoost) over all datasets for CMICOT with different $t = 1, \ldots, 10$.

Figure E.4: The average ranks of compared FS methods w.r.t. (M)AUC for $k = 1, \ldots, 50$ and for each of three classification models (NBC, kNN, and AdaBoost) over all datasets for all baselines and CMICOT with $t = 6$.

### E.1.2 Aggregated results over all $k$ and all dataset

In Table E.1, Table E.2

Table E.1: Aggregated (M)AUC, averaged over the datasets and $k = 1..50$.

| mAUC | NBC | kNN | AdaBoost |
|---|---|---|---|
| CMI | 0.777 | 0.776 | 0.851 |
| CMICOT ($t = 6$) | 0.775 | **0.815** | **0.866** |
| CMIM | **0.786** | 0.803 | 0.862 |
| DISR | 0.775 | 0.792 | 0.854 |
| FCBF | 0.783 | 0.752 | 0.801 |
| IWFS | 0.778 | 0.796 | 0.856 |
| JMI | 0.778 | 0.795 | 0.853 |
| MIFS | 0.776 | 0.774 | 0.832 |
| mRMR | 0.782 | 0.774 | 0.841 |
| RCDFS | 0.773 | 0.796 | 0.856 |
| RelaxMRMR | **0.786** | 0.779 | 0.845 |

Table E.2: Aggregated method rank, averaged over the datasets and $k = 1..50$.

| | NBC | kNN | AdaBoost |
|---|---|---|---|
| CMI | 6.8 | 7.4 | 6.4 |
| CMICOT ($t = 6$) | 6.3 | **3.5** | **4.2** |
| CMIM | 4.7 | 5.0 | 4.8 |
| DISR | 7.2 | 6.9 | 7.2 |
| FCBF | 4.7 | 6.6 | 6.5 |
| IWFS | 6.5 | 6.2 | 6.2 |
| JMI | 7.0 | 6.2 | 6.9 |
| MIFS | 6.0 | 5.9 | 5.7 |
| mRMR | 5.4 | 6.6 | 6.4 |
| RCDFS | 6.8 | 6.0 | 6.3 |
| RelaxMRMR | **4.4** | 5.6 | 5.3 |

## E.2 Results per each dataset

We apply 10-fold cross-validation to estimate the significance of differences in quality of FS algorithms for $k = 50$. In our protocol we launch a new feature selection before each training (so there are 10 FS rankings). Results are in Table E.6. We utilize Student's paired two-tailed t-test to evaluate the statistical significance of the difference between the mean scores.

We also compare the classifier quality at the optimal $k$ point, where the optimal $k \in 1...50$ is the number of features where the highest (M)AUC is achieved (see Tables E.3, E.4, and E.5). The average best M(AUC) obtained with CMICOT (with $t = 6$) feature rankings exceeds that of all other methods. Interestingly, according to the (M)AUC values, our strongest competitor is CMIM, not one of the three interaction-aware FS methods.

Table E.3: (M)AUC at the optimal $k$ (NBC).

| NBC | CMI | CMICOT ($t=6$) | CMIM | DISR | FCBF | IWFS | JMI | MIFS | mRMR | RCDFS | Relax-MRMR |
|---|---|---|---|---|---|---|---|---|---|---|---|
| arrhythmia | 0.588 | 0.579 | 0.619 | 0.626 | 0.631 | **0.650** | 0.638 | 0.630 | 0.650 | 0.648 | 0.648 |
| coil2000 | 0.690 | 0.685 | 0.681 | **0.705** | 0.679 | 0.697 | 0.688 | 0.698 | 0.694 | 0.687 | 0.696 |
| gizette | 0.903 | 0.922 | 0.947 | **0.954** | 0.947 | 0.907 | 0.933 | 0.940 | 0.953 | 0.894 | 0.949 |
| isolet | 0.976 | **0.994** | 0.991 | 0.980 | 0.996 | 0.991 | 0.976 | 0.989 | 0.988 | 0.968 | 0.991 |
| libras | 0.705 | 0.699 | 0.665 | 0.667 | 0.701 | 0.699 | 0.659 | **0.750** | 0.748 | 0.677 | 0.747 |
| madelon | 0.667 | 0.684 | 0.691 | **0.692** | 0.648 | 0.681 | 0.691 | 0.657 | 0.672 | 0.682 | 0.671 |
| poker2 | 0.508 | 0.509 | **0.512** | 0.503 | 0.508 | 0.503 | 0.503 | 0.508 | 0.508 | 0.503 | 0.508 |
| ranking | 0.816 | 0.818 | 0.810 | 0.803 | 0.798 | 0.813 | 0.812 | 0.807 | 0.816 | 0.814 | **0.819** |
| semeion | 0.901 | 0.966 | **0.975** | 0.936 | 0.933 | 0.905 | 0.948 | 0.957 | 0.953 | 0.917 | 0.960 |
| smartphone | 0.973 | 0.973 | **0.984** | 0.959 | 0.979 | 0.979 | 0.960 | 0.982 | 0.966 | 0.980 | 0.967 |
| usps | 0.955 | **0.986** | 0.984 | 0.970 | 0.987 | 0.965 | 0.981 | 0.979 | 0.982 | 0.966 | 0.984 |
| # times wins | 0 | 2 | **3** | **3** | 0 | 1 | 0 | 1 | 0 | 0 | 1 |

Table E.4: (M)AUC at the optimal $k$ (kNN).

| kNN | CMI | CMICOT ($t=6$) | CMIM | DISR | FCBF | IWFS | JMI | MIFS | mRMR | RCDFS | Relax-MRMR |
|---|---|---|---|---|---|---|---|---|---|---|---|
| arrhythmia | 0.558 | **0.590** | 0.529 | 0.537 | 0.563 | 0.523 | 0.553 | 0.533 | 0.536 | 0.558 | 0.511 |
| coil2000 | 0.669 | 0.669 | 0.669 | 0.669 | 0.669 | 0.669 | 0.669 | 0.669 | 0.669 | 0.669 | 0.669 |
| gizette | 0.960 | **0.983** | 0.980 | 0.971 | 0.961 | 0.956 | 0.973 | 0.918 | 0.974 | 0.956 | 0.980 |
| isolet | 0.951 | **0.986** | 0.970 | 0.954 | **0.986** | 0.966 | 0.938 | 0.968 | 0.959 | 0.926 | 0.972 |
| libras | 0.934 | 0.947 | 0.937 | 0.937 | 0.681 | 0.932 | 0.948 | 0.920 | 0.927 | **0.954** | 0.929 |
| madelon | 0.856 | 0.935 | 0.937 | 0.943 | 0.639 | 0.942 | **0.944** | 0.657 | 0.739 | 0.939 | 0.726 |
| poker2 | 0.667 | **0.735** | 0.666 | 0.676 | 0.505 | 0.694 | 0.676 | 0.685 | 0.685 | 0.695 | 0.692 |
| ranking | 0.760 | 0.766 | 0.767 | **0.774** | 0.752 | 0.764 | 0.763 | 0.754 | 0.766 | 0.766 | 0.763 |
| semeion | 0.909 | 0.961 | 0.964 | 0.912 | 0.949 | 0.910 | 0.928 | **0.966** | 0.925 | 0.923 | 0.934 |
| smartphone | 0.920 | 0.969 | **0.970** | 0.944 | 0.963 | 0.963 | 0.926 | 0.964 | 0.922 | 0.954 | 0.937 |
| usps | 0.945 | 0.991 | 0.992 | 0.975 | **0.993** | 0.978 | 0.982 | 0.989 | 0.982 | 0.984 | 0.988 |
| # times wins | 0 | **4** | 1 | 1 | 2 | 0 | 1 | 1 | 0 | 1 | 0 |

Table E.5: (M)AUC at the optimal $k$ (AdaBoost).

| AdaBoost | CMI | CMICOT ($t=6$) | CMIM | DISR | FCBF | IWFS | JMI | MIFS | mRMR | RCDFS | Relax-MRMR |
|---|---|---|---|---|---|---|---|---|---|---|---|
| arrhythmia | 0.578 | 0.664 | 0.639 | 0.647 | 0.661 | 0.658 | 0.653 | 0.661 | 0.637 | **0.675** | 0.646 |
| coil2000 | 0.712 | 0.711 | 0.726 | **0.758** | 0.698 | 0.716 | 0.712 | 0.714 | 0.755 | 0.712 | 0.741 |
| gizette | 0.979 | **0.998** | 0.995 | 0.985 | 0.985 | 0.974 | 0.988 | 0.946 | 0.992 | 0.978 | 0.993 |
| isolet | 0.992 | **0.996** | **0.996** | 0.992 | **0.996** | 0.995 | 0.986 | 0.995 | 0.992 | 0.986 | 0.994 |
| libras | **0.985** | 0.982 | 0.982 | 0.976 | 0.779 | 0.984 | 0.976 | 0.964 | 0.963 | 0.981 | 0.962 |
| madelon | 0.872 | 0.955 | 0.967 | 0.965 | 0.599 | **0.973** | 0.965 | 0.634 | 0.825 | 0.972 | 0.787 |
| poker2 | **0.857** | 0.840 | 0.789 | 0.851 | 0.508 | 0.825 | 0.851 | 0.840 | 0.840 | 0.827 | 0.830 |
| ranking | 0.830 | 0.828 | 0.823 | 0.817 | 0.811 | 0.825 | 0.824 | 0.818 | 0.823 | 0.830 | **0.834** |
| semeion | 0.970 | **0.991** | 0.989 | 0.958 | 0.981 | 0.957 | 0.965 | 0.989 | 0.969 | 0.961 | 0.970 |
| smartphone | 0.975 | 0.982 | **0.988** | 0.976 | 0.981 | 0.981 | 0.973 | 0.984 | 0.977 | 0.985 | 0.983 |
| usps | 0.970 | **0.998** | 0.996 | 0.989 | 0.997 | 0.989 | 0.993 | 0.997 | 0.996 | 0.992 | 0.996 |
| # times wins | 2 | **4** | 2 | 1 | 1 | 1 | 0 | 0 | 0 | 1 | 1 |

Table E.6: (M)AUC, mean difference between CMICOT (with $t = 6$) and the baseline methods, calculated using *full* 10-fold CV protocol on top 50 features over three representative datasets. Gray values indicate statistically insignificant results ($p\text{-}value > 0.05$).

| | NBC | | |
| --- | --- | --- | --- |
| | poker | semeion | usps |
| CMI | 0 | 0.048 | 0.026 |
| CMIM | -0.002 | 0.002 | 0.003 |
| DISR | 0.003 | 0.040 | 0.020 |
| FCBF | -0.005 | 0.024 | 0.003 |
| IWFS | -0.004 | 0.041 | 0.022 |
| JMI | 0.004 | 0.040 | 0.008 |
| MIFS | -0.002 | 0.004 | 0.051 |
| mRMR | -0.002 | 0.037 | 0.007 |
| RCDFS | 0 | 0.039 | 0.021 |
| RelaxMRMR | -0.003 | 0.025 | 0.004 |

| | kNN | | |
| --- | --- | --- | --- |
| | poker | semeion | usps |
| CMI | 0.122 | 0.051 | 0.049 |
| CMIM | 0.113 | -0.004 | 0.004 |
| DISR | 0.051 | 0.039 | 0.016 |
| FCBF | 0.282 | 0.013 | 0.005 |
| IWFS | 0.066 | 0.047 | 0.016 |
| JMI | 0.051 | 0.040 | 0.010 |
| MIFS | 0.107 | -0.006 | 0.005 |
| mRMR | 0.105 | 0.032 | 0.009 |
| RCDFS | 0.061 | 0.047 | 0.013 |
| RelaxMRMR | 0.092 | 0.022 | 0.007 |

| | AdaBoost | | |
| --- | --- | --- | --- |
| | poker | semeion | usps |
| CMI | 0.014 | 0.022 | 0.022 |
| CMIM | 0.062 | -0.002 | 0.001 |
| DISR | 0.001 | 0.021 | 0.008 |
| FCBF | 0.350 | 0.006 | 0.001 |
| IWFS | 0.038 | 0.020 | 0.008 |
| JMI | -0.002 | 0.018 | 0.004 |
| MIFS | 0.037 | -0.003 | 0.001 |
| mRMR | 0.033 | 0.016 | 0.002 |
| RCDFS | 0.02 | 0.021 | 0.006 |
| RelaxMRMR | 0.043 | 0.010 | 0.002 |

# F   Other additional experiments

## F.1   Comparison of the order of selected features

Figure F.1: Similarity between CMICOT (with $t = 6$) and the baselines in terms of the order of selected features measured by Kendall tau rank correlation coefficient Taylor (1987).

In this experiment we evaluate the difference between the order in which features were selected by one of baseline FS methods and the one of CMICOT (with $t = 6$). For that purpose we calculate a Kendall's tau measure Taylor (1987) of similarity between each baseline order ranking and CMICOT's one for top $k$ selected features (we consider $k = 10, 20, 50$). We do it for each dataset and, then, calculate an average score over those datasets. The results are shown in Figure F.1.

We see that the top-10 subsets of CMICOT are surprisingly close to those of RCDFS, which is another interaction-aware filter. There is some degree of similarity to RelaxMRMR as well. Concerning the groups (b) and (c) of our baselines, the best match is CMI, which is somewhat encouraging, since the method CMI is the idealistic but practically infeasible (for large $k$, it requires a drastically large number of instances to have accurate estimation of MIs in its score, see Sec.2 and Brown et al. (2012)). The closeness of FS methods to the "true" CMI method (i.e., as if it was calculated based on known joint distributions of all features) is studied in Appendix F.2.

## F.2 Comparison with the "true" CMI score

As we noted in Section 2, the core intuition of most existing filters based on SFS and conditional mutual information is to select features $f \in F \setminus S_{i-1}$ with the largest value of a low-dimensional approximation $J_i(\cdot)$ of the CMI score $J_i^{\mathrm{CMI}}(f) = \mathrm{I}(c; f \mid S_{i-1})$, since, for large $k$, it requires a drastically large number of instances to have accurate estimation of MIs in its score, see Sec.2 and Brown et al. (2012). Hence, one can conduct an experiment to evaluate the FS method with the score $J_i(\cdot)$ in terms the closeness of the score $J_i(\cdot)$ to the $J_i^{\mathrm{CMI}}(\cdot)$.

Figure F.2: Comparison of FS methods in terms of closeness of $\mathrm{I}(c; S_k)$ (where top-k features $S_k$ are selected by means of $J_{i,\mathcal{D}'}(f)$ of a compared FS method) to the true $\mathrm{I}(c; S_k)$ (where top-k features $S_k$ are selected by means of the true score $J_{i,\mathbb{P}_\mathcal{D}}^{\mathrm{CMI}}(f)$ of a compared FS method). We present results for two sample dataset sizes: 100 instances and 1000 instances. CMICOT is with $t = 6$.

Thus, we are faced to the problem of calculating the "true" score of CMI method, i.e., as if it was calculated based on known joint distributions of all features. The general assumption of machine learning is that the available dataset $\mathcal{D}$ of observations (comprising both train and test sets) is only a small sample of the same (latent) joint distribution $\mathbb{P}$ of features $f \in F$ and label $c$. In this setting, we cannot evaluate the true value of $J_i^{\mathrm{CMI}}(f) = J_{i,\mathbb{P}}^{\mathrm{CMI}}(f)$ to compare with the score $J_i(\cdot)$ of the evaluated method, since we don't have the complete information about distribution $\mathbb{P}$ underlying $J_{i,\mathbb{P}}^{\mathrm{CMI}}(f)$. However, we can mimic this setting with the following idea of scaling down the task. Namely, we substitute dataset $\mathcal{D}$ with its bootstrap subsample $\mathcal{D}'$ and consider the empirical distribution $\mathbb{P}_\mathcal{D}$ defined by $\mathcal{D}$ instead of the actual latent distribution $\mathbb{P}$. In this way, we can exactly evaluate $J_{i,\mathbb{P}_\mathcal{D}}^{\mathrm{CMI}}(f)$ and evaluate closeness of the score $J_{i,\mathcal{D}'}(f)$ calculated over the smaller dataset $\mathcal{D}'$ to $J_{i,\mathbb{P}_\mathcal{D}}^{\mathrm{CMI}}(f)$, and, thus reveal its ability to recover the ordering of features $f \in F \setminus S_{i-1}$.

We embody those considerations with a simple procedure. We run feature selection on several samples of a chosen dataset (i.e., poker). The consider sample size equal to either 100 or 1000 instances, the number of bootstrap replications for each dataset is 30. We picked 6 baselines to compare with: all the group (a) (it contains interaction-aware methods); and CMI that is estimated by $J_{i,\mathcal{D}'}^{\mathrm{CMI}}(f)$ (i.e., it approximate the true CMI score $J_{i,\mathbb{P}_\mathcal{D}}^{\mathrm{CMI}}(f)$). We consider $k = 30$ and, for each evaluated FS method, we calculate the MI $\mathrm{I}(c; S_k)$ (where top-k features $S_k$ are selected by $J_{i,\mathcal{D}'}(f)$) for each sample, then average it across the replications. We use $\mathrm{I}(c; S_k)$, calculated on the top $k$ features selected by CMI on the full dataset (i.e., by means of the true score $J_{i,\mathbb{P}_\mathcal{D}}^{\mathrm{CMI}}(f)$) as a reference value of $\mathrm{I}(c; S_k)$, to which all the sample MIs converge. In this way, the full dataset simulates the "general population", while

the results of CMI run on the full dataset approximate the "ground truth" ranking. The estimated mean scores with confidence intervals are presented in Figure F.2.

Indeed, as was earlier noticed by Brown et al. (2012), CMI performs drastically worse on small datasets, which is not the case with our approach (the difference between $k = 100$ and $k = 1000$ is insignificant) and other baselines. CMICOT shows closest MI of selected features to the reference MIs obtained on a full dataset.

## Footnotes

[1]E.g., we can consider 3 jointly i.i.d. Bernoulli random variables $f_1, f_2, f_3$ with success probability $1/2$. Then, let $f_0 = f_1$, $c = f_1 + f_2 (\mod 2)$.

[2]One can show that this set will be selected by the algorithm in priority to $f_0$ (in the following order: $f_3, f_1, f_2$).

[3] http://archive.ics.uci.edu/ml/datasets.html