[Reviews · NeurIPS 2016]

Reviewer 1

Summary

This paper presents a new feature selection algorithm, named CMICOT, that considers high-order dependences among features (up to order t, with t > 3), and uses Mutual Information (MI) to measure such dependences. Additionally, in order to alleviate the lack of data and the increase on computational cost to properly estimate the mutual information, the CMICOT algorithm is a greedy approach that uses binary representatives of each feature. The proposed algorithm is tested against some state-of-the-art feature selection algorithms, and interaction-aware SFS-based filters. The presented results on 10 public datasets show that overall the CMICOT algorithm outperforms the other feature selection algorithm in terms of classification accuracy when kNN or AdaBoost are used as classifiers.

Qualitative Assessment

The problem of designing a feature selection algorithm capable of efficiently deal with high-order interaction among features is an interesting and open problem in the feature selection area. That is why this paper is appealing. However, there are several issues regarding the computational cost and the experimental setup that need a clarification in order to consider it for acceptance. • My main concern is about the justification and analysis of the Binary representatives procedure described in Section 3.3, which I think compromises the acceptance of this paper. • On the one and, the theoretical justification of this procedure is not very convincing. It is said (lines 219-222) that “The described technique has been inspired by the intuition that probably two binary representatives of two different features interact on average better than two binary representatives of one feature”; however, no references or examples are provided to support this idea. On the other hand, when comparing the computational cost between the algorithm with and without binary representations (lines 215-219), the same values for t and s are considered. This is not a fair comparison as both cases are not taking the same level of information. Additionally, I have serious doubts about the reduction in computational cost of the binary representatives. Then, in Algorithm 2, when searching for complementary features and opposing features, I can see a reduction in the cost of computing the mutual information (we are working with binary variables), but the cardinalities of S^bin U B[f] and S^bin are larger than in the non-binary case, as all the set B[f_best] is added to the set S^bin set in line 21. In short, it seems like there is a trade-off between the computational cost to compute the argmax and the mutual information. • Lines 180-182: If an “optimal” interaction-aware MI-based feature selection method is wanted and not a low dimensional approximation of G, the proposed method as a computational cost O(i^2), which is the same as RelaxMRMR. What are reasonable values for t and s? Please, clarify this point. • Please, rephrase/clarify lines 183-191. • I don’t understand why the particular case t=s is described in Algorithm 1, and a generic algorithm is not provided. • Proposition 3. First sentence in Appendix A.4 needs clarification and it is fundamental for the correctness of Proposition 3 (“Proof. The calculation of a joint entropy of m variables over N requires takes O(mN) simple operations. Hence, any MI that involve m variables requires O(mN) simple operations as well.”). • Line 249-250: “but estimation of…. BR technique.” Please clarify this sentence. Isn’t it a problem-dependent factor? • Experimental setup. There is another critical point here. It is not clear whether the curves presented in Figure 1 are obtained over a validation/test set or over the training set. It is said later on that 10-fold cross-validation is applied to estimate the significance of differences in classification quality, but it is not clear whether this procedure was also applied in Figure 1.This must be clarified. • The proposed method does not provide good results for the NBC classifier; though these results are provided in the appendices, they are not mentioned or even discussed in the main paper. • There is relevant information in the appendices regarding the comparison between different feature selection methods. I think that the average results reported in the main manuscript are not enough. • Experiments. How can it be explained that the strongest competitor is CMIM, and not one of the interaction-aware FS methods? • It would be interesting to empirically compare CMICOT with and without binary representatives in terms of classification accuracy and computational cost. Additionally, it is highly advisable to include some comparisons with the other feature selection methods in terms of computational cost. It is especially critical to make it clear the advantage of the proposed method with respect to other interaction-aware SFS-based filters. • Minor comments: - Line 27: Should it be SFS-based filters instead of SBS-based filters? - Lines 144-145: clarify the sentence in cursive. - Missing axes labels in Figure 1. - Please, indicate the statistical test used in Table 1.

Confidence in this Review

3-Expert (read the paper in detail, know the area, quite certain of my opinion)


Reviewer 2

Summary

This paper describes an information theoretic feature selection algorithm which approximates a high order mutual information by decomposing categorical variables into binary variables. It further decomposes the high order mutual information into a search over two sets, one which finds the best complementary subset (in terms of increasing the information), and another which finds the subset that contains most of that information.

Qualitative Assessment

Major comments: Statement 1 and the associated proof is unnecessary. When t & s are greater than the number of selected features then the maximum score that can be assigned to a variable is the mutual information between it combined with all the selected features against the target. Either all variables end up in H or they end up in G depending on how the search is done. This score is then rank equivalent to the CMI score. The behaviour of this criterion is only interesting when it's actually approximating the CMI (i.e. when t & s < i). The interplay between the max and min steps is very interesting. I would expect that most of the time G covers H, as this would minimise the information, but in the case of complementary features then it becomes very dependent on the search procedure. The use of the binary representations blows up the search space and allows the algorithm to tease out interesting interactions. The complexity analysis of the competitor techniques is strictly correct that they increase in computation as i increases, but this is only with a naive implementation. With a memoized implementation (e.g. in the FEAST toolbox used for the experimental study) there are O(d) mutual information calculations in each iteration so the computation does not grow over time, with O(kd) memory required and O(kd) mutual information calculations total. In CMICOT it doesn't appear that these information calculations can be memoized (as the binary features chosen can change) so the computational complexity will grow over time as discussed in the paper. The experimental study lacks a few details (k in k-NN, which multiclass Adaboost is used, what is the base learner for Adaboost and how many ensemble members). The notation in portions of the supplementary material is unclear. For example in the equation above line 21 each of the Gs is a different subset, so they should use different letters (or G' etc). Much of the detail relies on the fact that H and G can have a large intersection (or even H \cup G) which essentially removes H from the mutual information. Minor comments: The proof of proposition 3 doesn't define a simple operation. Calculating a joint entropy requires iterating over all N datapoints and then over all the states of the joint variable, but this gives complexity O(N + |X|) rather than O(Nm) where |X| is the number of states. Separating out the results for poker, ranking and semeion would allow the reader to separate the binary approximation from the max/min criterion. Does the binary approximation improve performance by adding some noise or reduce performance by limiting the amount of information as compared to JMI or RelaxMRMR? With separate results for binary features vs non-binary features it might be possible to answer that question. As part of this it would be interesting to run all the other algorithms on the binary expanded versions of the datasets to see how that affected the performance.

Confidence in this Review

3-Expert (read the paper in detail, know the area, quite certain of my opinion)


Reviewer 3

Summary

The paper proposed a sequential forward selection style feature selection method which can identify high-order feature interactions. The authors also proposed binary representatives of features which makes their feature selection approach scalable. Empirical comparison is provided to support its superior performance.

Qualitative Assessment

[Technical quality] The experimental methods are appropriate, but more empirical experiments can be added. Also I expect more analysis over the experiments. [Novelty/originality] The methodology contained in this paper is incremental. [Potential impact or usefulness] The potential impact of the work is not particular influential, but considering high-order interaction in feature selection is an important research question. [Clarity and presentation] I think the overall presentation of the paper is ok, but the notations can be misleading in some places. Although the authors states in Footnote 2 that I(f:g,h):=I(f;(g,h)), but I feel this abbreviation can lead to misunderstanding in many places. For example, in Section 3.1, it is much better to use I(c;{f,H}) rather than I(c;f,H). Also in their footnote 2, F={f_i}_{i=1}^{n} should be F=\cup_{i=1}^{n}{f_i}. The authors did a great job summarizing related work, but the following highly related work is missing. High-dimensional structured feature screening using binary Markov random fields. by Liu et al 2012 [Qualitative assessment] Rewriting the paper, especially improving the notations can be quite helpful. Adding more experiments is a good idea since the current empirical experiments are not intensive. I also wonder whether the authors can further explore whether submodularity can be used to understand their approach's theoretical behavior.

Confidence in this Review

3-Expert (read the paper in detail, know the area, quite certain of my opinion)


Reviewer 4

Summary

The paper introduces a feature selection algorithm, which claims the following contributions: 1) apply a two stage greedy search algorithm to MI. 2) utilization of binary representation of features They name their method as CMICOT. In comparison with various practical experiments, CMICOT demonstrates better performance than different related methods.

Qualitative Assessment

Several problems in this paper: 1) line 27, what is SBS based filters? 2) what's the y-axis of figure 1? Since this is the only experiment figure demonstrated in this paper, lacking of y axis makes me difficult in understanding the effectiveness of this work. Sometimes, these figures may represent a significant difference of 10, while it can also be negligible, such as 0.00001.

Confidence in this Review

2-Confident (read it all; understood it all reasonably well)


Reviewer 5

Summary

Authors propose CMICOT, a filter-based feature selection method that is able to discover interactions between more than 3 features -- which is a common bottleneck in the higher orders able to be captured.

Qualitative Assessment

Submission reads very cleanly and the authors build their argument for the proposed approach well placing it clearly with respect to the general challenges of the domain and existing literature. Formalization of the approaches are provided. However, this work is a little light in evaluation. Although a number of datasets are presented performance is aggregated to the point identifying how (i.e., magnitude of outperformance) and why (underlying reasons for the benefits) are unclear. Additional results seen in the appendix but the material presented within the page limit is, in of its self, lacking. Figure 1 in particular does not have a y-axis to interpret the relative performance gaps.

Confidence in this Review

1-Less confident (might not have understood significant parts)